# Newborn *MTHFR* rs1801133 Variant and Extremely Low Birth Weight: A Case–Control Study and Meta-Analysis

**DOI:** 10.3390/genes16101192

**Published:** 2025-10-13

**Authors:** Bartosz Skulimowski, Anna Durska, Alicja Sobaniec, Anna Gotz-Więckowska, Ewa Strauss

**Affiliations:** 1Department of Ophthalmology, Poznan University of Medical Sciences, 60-569 Poznan, Poland; skulimowskibartosz@gmail.com (B.S.); agotzwieckowska@ump.edu.pl (A.G.-W.); 2Institute of Human Genetics of the Polish Academy of Sciences, 60-479 Poznan, Poland; anna.durska@igcz.poznan.pl; 3Department of Neonatology, Gynecology and Obstetrics Clinical Hospital, Poznan University of Medical Sciences, 60-535 Poznan, Poland; alicja.sobaniec@gmail.com

**Keywords:** extremely low birth weight (ELBW), extremely low gestational age (ELGA), prematurity, MTHFR, PON1, homocysteine metabolism, genetic variants, fetal genotype, bronchopulmonary dysplasia (BPD), patent ductus arteriosus (PDA)

## Abstract

**Background:** Extremely low birth weight (ELBW) and extremely low gestational age (ELGA) remain major challenges in neonatology, contributing to neonatal morbidity and mortality. This study aims to examine the association between functional variants of *MTHFR* and *PON1*, genes involved in homocysteine metabolism, and the risk of ELGA, ELBW, and other complications of prematurity. A meta-analysis was also conducted to integrate literature data with the results of this study. **Methods:** The study included 377 premature infants, 164 mothers, and a population-based sample of 404 individuals. Genotyping was performed using TaqMan assays. **Results:** The fetal, but not maternal, *MTHFR* rs1801133 genotype was associated with ELBW (OR = 1.65; 95% CI: 1.09–2.51; *p* = 0.017, dominant model), bronchopulmonary dysplasia (*p* = 0.028), patent ductus arteriosus (*p* = 0.017), and neonatal mortality. The meta-analysis, which included five studies spanning 1156 cases and 1124 controls, confirmed the association between the neonatal *MTHFR* genotype and low birth weight (LBW), demonstrating an association of the rs1801133T allele with LBW in the TT homozygote model (vs. CT: OR = 1.41; 95% CI: 1.08–1.80; *p* = 0.0097). Subgroup analyses indicated that the rs1801133T allele is a protective factor against LBW in more developed countries, such as Canada and the UK (dominant model), whereas in other countries, such as China, Turkey, and Poland, it is a risk factor for LBW (recessive model). No association with *PON1* variants with ELBW or ELGA was found. **Conclusions:** This study provides the first global evidence confirming that the neonatal *MTHFR* genotype contributes to LBW, underscoring the population-specific effects of this genetic variant.

## 1. Introduction

Advances in neonatology and obstetric care have significantly improved the survival of infants born prematurely, with low birth weight (LBW, <2500 g), or small for gestational age (SGA). It is estimated that one-quarter of liveborn infants are now affected by one or more of these three vulnerabilities. Because of their similar pathogenic mechanisms and therapeutic approaches, these newborns have recently been collectively termed small vulnerable newborns (SVNs) [1]. Among SVNs, those with extremely low birth weight (ELBW), defined as a birth weight below 1000 g, and extremely low gestational age (ELGA), characterized as a gestational age below 28 weeks, represent the phenotypes of extreme prematurity. These infants remain at the highest risk for prematurity-related morbidity and perinatal mortality and require the most intensive and specialized medical care [2,3].

ELBW and ELGA are multifactorial traits influenced by both maternal and fetal factors [4,5]. The genetic basis of these traits remains incompletely understood. Twin concordance studies suggest that maternal genetic factors have a significant influence on gestational age, a pattern consistent with mitochondrial inheritance [6,7]. By contrast, variation in birth weight is determined mainly by fetal genes, with heritability estimated at around 70% [8]. Recent genome-wide association studies have identified 60 variants associated with birth weight, showing that the fetal genotype accounts for most of the variability in birth weight in 93% of the identified loci [9]. Among the candidate genes, the methylenetetrahydrofolate reductase (*MTHFR*) gene is one of the most frequently analyzed in the context of low gestational age and low birth weight. Previous studies, summarized in meta-analyses, have confirmed the important role of the maternal *MTHFR* genotype [10,11]. However, few studies have been conducted to date to investigate the role of fetal genotype, and none have specifically addressed its association with ELBW or ELGA.

MTHFR is an enzyme involved in the metabolism of folate, which catalyzes the transfer of methyl groups necessary for DNA synthesis, methylation, and the conversion of the toxic metabolite homocysteine (Hcy) to methionine [12]. The *MTHFR* gene is located on chromosome 1 [1p36.22] and has a common variant rs1801133 (c.665C>T; p.Ala222Val) resulting in reduced enzyme stability and activity [13]. In rs1801133 CT heterozygotes, enzyme activity is about 64% of normal, while in TT homozygotes it decreases to ~32%. This reduction in MTHFR activity leads to elevated Hcy levels, particularly when folate intake is insufficient [14,15]. Given the crucial role of methyl groups in fetal growth and development [16,17], supplementation with folic acid or its active form, 5-methyltetrahydrofolate, is recommended both before and during pregnancy [18,19].

The toxic effects of elevated Hcy levels are exerted in part through an increase in the concentration of its reactive derivative, Hcy-thioxanthone (Hcy-TLC) [20]. This compound is detoxified by paraoxonase 1 (PON1) [21], an antioxidant enzyme encoded by the *PON1* gene located on chromosome 7 [7q21.3]. Two common coding variants of *PON1*, rs854560 (c.163T>A; p.Leu55Met) and rs662 (c.575A>G; p.Gln192Arg), influence hydrolytic activity toward Hcy-TLC (paraoxonase/thiolactonase). The rs854560A (55Met) allele is associated with lower thiolactonase activity, while the rs662G (192Arg) allele is associated with higher thiolactonase activity, with rs662GG homozygotes showing about four-fold-higher activity compared to rs662AA individuals [22,23]. Moreover, the presence of the rs854560A allele is associated with elevated serum PON1 concentrations, which can be attributed in part to linkage disequilibrium with certain promoter variants that influence gene expression [24,25].

This study aims to comprehensively evaluate the relationship between functional variants of *MTHFR* and *PON1*, key genes involved in Hcy and Hcy–TLC metabolism, and the risk of ELGA, ELBW, and other complications of prematurity. A meta-analysis integrating literature data with the study’s results was also performed to validate the impact of the *MTHFR* variant on birth weight.

## 2. Materials and Methods

### 2.1. Study Population

The study included a population of 377 infants born before 32 weeks of gestation enrolled between 2009 and 2020 from the Neonatal Intensive Care Unit at the Clinical Hospital of Gynecology and Obstetrics at Poznan University of Medical Sciences. In the first stage of the study (between 2009 and 2014), 164 mothers of premature infants were also recruited. Additionally, 404 individuals (49.9% women) with mean age 49.1 ± 15.5 years from the local adult population (Great Poland region) without known signs of chronic diseases were analyzed to obtain population genotype frequencies (see Figure 1 for details). The exclusion criteria for the study included infants born from multiple pregnancies and those with chromosomal abnormalities. All participants in the study were of Caucasian origin.

### 2.2. Clinical Features and Outcomes

The following clinical factors were analyzed: gestational age (weeks), birth weight (grams), sex, preterm prelabor rupture of membranes (PPROM) duration (days), mode of delivery (vaginal delivery vs. cesarean section), Apgar scores at 1 and 5 min, blood transfusions, surfactant and oxygen therapy, and duration of mechanical ventilation (days). ELGA was diagnosed for gestational age <28 weeks and ELBW for birth weight <1000 g. The incidence of SGA and fetal growth restriction (FGR; formerly termed intrauterine growth retardation, IUGR) was assessed. The diagnosis of SGA was based on WHO criteria, defined as neonatal weight below the 10th percentile for gestational age [26]. FGR was diagnosed according to Lees et al. (2020), based on biometric and Doppler ultrasonography, with consideration of gestational age and growth dynamics [27]. The presence of BPD, intraventricular hemorrhage (IVH), patent ductus arteriosus (PDA), respiratory distress syndrome (RDS), necrotizing enterocolitis (NEC), periventricular leukomalacia (PVL), and retinopathy of prematurity (ROP), including its proliferative type (pROP), was assessed. The criteria for diagnosing BPD, IVH, RDS, NEC, PVL, and ROP have been described in detail previously [28]. The diagnosis of PDA was based on ultrasonography.

### 2.3. Genotyping

For DNA isolation, 1 ml of venous blood or a swab from the inner cheek (in duplicate) was collected. The samples were frozen at −80 °C and delivered to the Institute of Human Genetics of the Polish Academy of Sciences, where genomic DNA isolation and genetic analysis were performed. Genomic DNA was extracted from circulating blood lymphocytes using the QIAamp DNA Kit (Qiagen GmbH, Hilden, Germany) or from buccal swabs using the innuPREP DNA Kit (Analytik Jena AG, Jena, Germany), according to the manufacturer’s instructions. Analysis of the three variants, *MTHFR* rs1801133, *PON1* rs854560, and rs662, was performed using quantitative polymerase chain reaction (qPCR) with commercially available, pre-designed TaqMan genotyping assays (Thermo Fisher Scientific, Waltham, MA, USA): C___1202883_20, C___2259750_20, and C___2548962_20, respectively). Genotype analysis was performed on an ABI 7900HT Fast Real-Time PCR System (Life Technologies, Carlsbad, CA, USA). Genotyping success rate was between 99–100%.

### 2.4. Ethical Statement

All procedures carried out on human participants in this study were in accordance with the ethical standards of the institutional and national research committee and with the 1964 Helsinki declaration and its later amendments (or comparable ethical standards). The study was approved by the Bioethics Committee of Poznan University of Medical Sciences (no. 1140/05, issued on 8 September 2005; no. 1117/18 from 7 November 2018). Written prior informed consent was obtained from all participants, including consent from the parents or guardians of the study infants.

### 2.5. Meta-Analysis of the MTHFR rs1801133 Variant and Risk of Low Birth Weight

The meta-analysis was conducted following the Preferred Reporting Items for Systematic Reviews and Meta-Analyses (PRISMA) statement [29]. Associations of the *MTHFR* rs1801133 variant with growth-related neonatal outcomes were examined.

#### 2.5.1. Search Strategy

A literature search was conducted in PubMed (including MEDLINE), Cochrane Library, Embase, Scopus, Web of Science, Google Scholar, and associated tools (including LitVar2 NCBI) to identify relevant studies on the neonatal *MTHFR* genotype. The search included publications up to 10 August 2025. A combination of the following keywords and MeSH terms was used: (“Methylenetetrahydrofolate reductase” OR “MTHFR” OR “C677T” OR “C665T” OR “rs1801133”) AND (“variant” OR “variation” OR “polymorphism” OR “mutation”) AND (“Preterm Birth” OR “Low Birth Weight” OR “Fetal Growth Retardation” OR “Intrauterine Growth Retardation” OR “Small for Gestational Age”). References contained in the retrieved articles were also reviewed and included as additional studies when they met the eligibility criteria. For studies using overlapping data or the same study group, only the most complete or recent studies were included.

#### 2.5.2. Inclusion and Exclusion Criteria

The following criteria were applied to include studies in the meta-analysis: (a) studies with a case–control, cohort or family-based design; (b) studies focusing on the association between *MTHFR* variant and LBW/FGR/IUGR/SGA; and (c) availability of sufficient genotype data in the case and control groups to calculate crude odds ratios (ORs) and 95% confidence intervals (CIs). The exclusion criteria were as follows: (a) studies not designed as case–control, cohort, or family-based; (b) lack of genotype data or inability to calculate it; (c) studies based on pedigree data, twins studies, and linkage studies; (d) case reports, review articles, posters, abstracts, and animal studies; and (e) studies with incomplete or overlapping data. In cases of overlapping or duplicate publications, only the largest or most recent dataset was included.

#### 2.5.3. Data Extraction

Two authors independently reviewed and extracted the following information from all included studies using a structured data collection form: first author name, publication date, participant location, ethnicity, presence of premature neonates, studied outcomes, genotyping method, sample sizes of cases and controls, genotype frequency distribution, minor allele frequency (MAF), and Hardy–Weinberg equilibrium (HWE) in controls. Any discrepancies were resolved by consensus among all authors. If detailed genotype or HWE information was not reported, we calculated the necessary data and added the relevant details.

#### 2.5.4. Quality Assessment

The quality (Q) of studies was independently assessed by the Newcastle–Ottawa quality scale (NOS) [30,31,32]. Studies with scores of 0–3, 4–6, 7–9 were, respectively, considered as low-, moderate-, and high-quality.

### 2.6. Statistical Analysis

Testing for HWE was performed using a *χ*^2^ test with the online tool available at https://www.had2know.org/academics/hardy-weinberg-equilibrium-calculator-2-alleles.html (accessed on 23 June 2025). For each variant, the MAF was calculated. The effect of genotypes was estimated using ORs with 95% CIs, with minor alleles considered as risk factors. Continuous variables were tested for normality using the Kolmogorov–Smirnov test. In statistical analyses, qualitative variables were analyzed using the *χ*^2^ test or Fisher’s exact test, while quantitative variables were assessed using the *t*-test. Variables deviating from the normal distribution were evaluated using Mann–Whitney U test.

The meta-analysis was performed using the online tool METAGENYO available at http://metagenyo.genyo.es/ (accessed on 11 August 2025) [33]. Six genetic models were examined for each variant: allele contrast, recessive, dominant, overdominant, homozygote comparison, and heterozygote comparison. Heterogeneity was evaluated using Cochran’s Q and I^2^ statistics [34], with FEM applied in cases of low heterogeneity and REM otherwise. For each model, ORs, *p*-values (95% CIs), and adjusted *p*-values were calculated; HWE in controls was tested with corresponding *p*-values and adjusted *p*-values. Publication bias was assessed using funnel plots and Egger’s test.

All reported tests were two-sided, with statistical significance defined as *p* < 0.05. Statistical analyses were performed using STATISTICA version 10.0 and GraphPad Prism version 6.04. Post hoc power analysis for associations was performed using Quanto v 1.2 software. For visualization of birth weight and gestational age distributions, a plot was generated using the “ggplot2” and “introdataviz” packages in the R environment version 4.5.1. [35,36,37].

## 3. Results

### 3.1. Demographic and Clinical Characteristics of the Study Group and Associations with ELBW and ELGA

The characteristics of the study cohort of 377 premature infants are presented in Table 1. We identified 149 infants with ELBW (39.5%) and 152 with ELGA (40.3%).

Demographic and clinical characteristics of ELBW and ELGA infants were compared with those of non-ELBW and non-ELGA infants to assess associations with extreme prematurity phenotypes. Both ELBW and ELGA infants had lower birth weight and gestational age. ELBW was positively associated with SGA (OR = 2.3; *p* = 0.0004) and FGR/IUGR (OR = 13.0; *p* < 0.0001), whereas ELGA was negatively associated with SGA (OR = 0.22; *p* < 0.0001) and showed a nonsignificant trend toward lower incidence of FGR/IUGR. As expected, ELBW and ELGA were associated with lower Apgar scores at 1 and 5 min, respiratory failure (including more frequent surfactant therapy, resuscitation, mechanical ventilation, and prolonged oxygen supplementation), increased incidence of prematurity-related complications (except PDA, which was associated with ELBW only), and higher mortality rates. No associations with sex were observed.

In the analysis of maternal risk factors, maternal age (mean 28.3 ± 5.8 years; range 17–45) was not associated with either ELBW or ELGA. The women studied had a mean of 1.3 ± 1.8 previous pregnancies, including 24 women (14.6%) with a history of preterm birth and 28 (17.1%) with a history of spontaneous miscarriage. Delivery by cesarean section was negatively associated with ELGA (*p* ≤ 0.01), with no effect on ELBW, whereas PPROM and prolonged PPROM were not associated with extreme prematurity phenotypes.

When comparing the clinical profiles of cohorts A (*n* = 164) and B (*n* = 213) to assess the consistency of risk factors over time, we observed lower birth weight (*p* = 0.02) and a higher incidence of FGR/IUGR (*p* = 0.049) in the more recent cohort B, accompanied by more intensive respiratory support, including prolonged mechanical ventilation (*p* = 0.0001) and oxygen supplementation (*p* = 0.020; Appendix A). A higher incidence of RDS (*p* < 0.0001) and proliferative ROP (*p* = 0.041), a complication related to oxygen supplementation, was also noted in this cohort. Conversely, mechanical ventilation was less frequently initiated in this group, and the incidence of PPROM and prolonged PPROM decreased over the study period.

### 3.2. Frequencies of MTHFR and PON1 Variants in the Study Groups

The distribution of the studied *MTHFR* and *PON1* genotypes was consistent with the Hardy–Weinberg equilibrium (*p* > 0.05) in all studied groups: infants, mothers, and the population group (Table 2). There were no differences in the distribution of studied variants between groups: the frequency of the *MTHFR* rs1801133T allele was between 0.293 and 0.330, the *PON1* rs854560T allele between 0.314 and 0.378, and the *PON1* rs662G allele between 0.265 and 0.287. For genotypes, two effects were found: the decreased frequency of *MTHFR* rs1801133T allele carriers in infants compared to the population group (OR = 0.75; *p* = 0.046 (Table 2, Figure 2)), and higher frequency of the *PON1* rs854560TT homozygotes in infants compared to mothers: OR = 2.00; *p* = 0.029.

### 3.3. Associations of MTHFR and PON1 Genotypes with ELBW and ELGA

The significant association between infant *MTHFR* rs1801133 genotype and ELBW was observed in the studied population (Table 3 and Figure 2). In the combined analysis of both cohorts (A + B), carriers of the rs1801133T allele had a 1.65-fold increased risk of ELBW (*p* = 0.017). Although studied cohorts did not differ significantly in genotype distribution, the association of the rs1801133T allele with ELBW tended to be stronger for homozygotes (OR = 2.4; *p* = 0.107) in cohort A and for heterozygotes (OR = 2.0; *p* = 0.021) in cohort B. Analysis by sex revealed that the *MTHFR* genotype in newborns (heterozygotes) is a risk factor for both ELBW (OR = 1.89; *p* = 0.038) and ELGA (OR = 1.90; *p* = 0.034; Appendix A). No statistically significant association was observed between maternal genotypes and ELBW or ELGA. The distribution of birth weight and gestational age according to sex and *MTHFR* rs1801133 genotype is shown in Figure 3.

### 3.4. Associations of MTHFR and PON1 Genotypes with Neonatal Comorbidities and Mortality

The study was conducted on the entire group of premature infants (*n* = 377; Table 4). Two statistically significant associations were observed for the *MTHFR* rs1801133T variant: the CT heterozygous genotype was associated with a 1.67-fold-higher risk of developing BPD (vs. CC; *p* = 0.017), and the TT homozygous genotype was associated with a 2.19-fold-higher risk of developing PDA (recessive model; *p* = 0.028). A borderline statistically significant association was also observed between the *PON1* rs662AG heterozygous genotype and the risk of PDA (vs. AA homozygotes: OR = 1.80; *p* = 0.053). No significant associations were found between the genotypes studied and the occurrence of RDS, IVH, NEC, ROP, or pROP. No significant associations were also observed for mortality, most likely due to the limited sample size. However, carriers of the *MTHFR* rs1801133T allele were highly prevalent among the deceased infants (75% of the group), showing a trend toward a 3.22-fold-increased risk of death (*p* = 0.156). A similar trend was observed for carriers of the *PON1* rs662G allele (OR = 3.19; *p* = 0.151).

### 3.5. Meta-Analysis of the Effect of MTHFR Genotype on Low Birth Weight

A literature search retrieved 464 articles potentially addressing the association between the neonatal *MTHFR* rs1801133 genotype and low birth weight. After removing 189 duplicate records, 201 unique articles remained. Screening of titles and abstracts reduced this number to twelve studies deemed suitable for full-text evaluation. Two Chinese reports could not be retrieved and therefore ten full texts were assessed for eligibility. Two of them were excluded due to a wrong publication type (review and meta-analysis) and another four due to the absence of neonatal genotype data. Applying the predefined inclusion and exclusion criteria ultimately yielded four eligible studies, all with a case–control design, comprising a total of 1007 cases and 896 controls [38,39,40,41]. Combined with our data, the meta-analyses included a total of 1156 cases and 1124 controls (Figure 4).

Table 5 summarizes the main characteristics of the studies included in the meta-analysis. Research on the *MTHFR* rs1801133 variant covered diverse geographic regions and ethnicities, including Asian (China), Caucasian (the UK, Turkey, and Poland), and mixed-ethnicity (Canada) populations. In all studies, genotyping was performed in newborns, with case definitions based on anthropometric or birth weight thresholds: SGA (<10th percentile), LBW (<2500 g), or ELBW (<1000 g). Two investigations, Chen et al. (2004) [39] and the present study (2025), included only preterm infants, whereas the others enrolled either mixed preterm and term groups or exclusively term neonates.

Study sizes varied widely, from the smallest dataset of 55 cases and 55 controls [41] to the largest with 467 cases and 461 controls [38]. The frequency of the T allele ranged from 0.255 in the Turkish population to 0.604 in the Chinese population. All control groups adhered to Hardy–Weinberg equilibrium. Quality assessment using the NOS assigned scores of 6 to 8 across all included studies, consistent with moderate to high methodological rigor.

The pooled analysis of five studies demonstrated a significant association between the homozygous TT genotype of the *MTHFR* variant and an increased risk of low birth weight. The best-fitting model was comparison of TT vs. CT genotypes (OR = 1.40; 95% CI: 1.80–1.71; *p* = 0.0097; FEM; Figure 5a). In a recessive model (TT vs. CT + TT) this association was also significant in FEM; however moderate heterogeneity was observed in this model (I^2^ = 60.4%, Figure 5b).

We performed subgroup analyses and found that in the UK and Canada—the most developed countries included in this meta-analysis—the T allele in the dominant model was associated with a reduced risk of LBW (OR = 0.79; 95% CI: 0.63–0.99; *p* = 0.038; FEM). In contrast, in countries such as China and Turkey, the T allele was associated with increased risk (Figure 6). In this meta-analysis, Poland also clustered with countries showing increased risk for the *MTHFR* variant. The strongest association was observed in the recessive model (OR = 1.85; 95% CI: 1.34–2.53; *p* < 0.0001; FEM). A summary of the overall and subgroup meta-analyses is presented in Appendix A.

The funnel plots for the studied associations (Figure 7a–d) showed no asymmetry, suggesting the absence of publication bias in performed analyses. This was further supported by Egger’s test, which revealed no statistically significant regression intercepts for the studied models (with the exception of the subgroup of developed countries, where calculation was not possible).

## 4. Discussion

Despite advances in perinatal care, ELGA and ELBW remain serious challenges in neonatology, contributing to severe short- and long-term health complications, reduced quality of life, and increased mortality [42,43]. In this study, we investigated the role of functional variants in two candidate genes involved in Hcy and TLC-Hcy metabolism, *MTHFR* and *PON1*, in relation to ELBW, ELGA, and neonatal complications of prematurity. The genetic analysis was conducted in infants, mothers, and a population sample, as well as complemented by a meta-analysis of available studies, thus allowing us to position our findings within a broader clinical and genetic context.

In the clinical evaluation, we confirmed previously described maternal risk factors for preterm delivery, including PPROM, short interpregnancy interval, and history of prior preterm delivery [44,45,46]. Approximately 15% of mothers reported a history of preterm birth, which suggests persistence of chronic or recurrent risk factors across pregnancies. High frequency of delivery by cesarean section (50.9%) was also noted, consistent with earlier reports [47,48]. Over the study period, we observed a decline in the frequency and duration of PPROM, reflecting improvements in obstetric care. However, neonatal outcomes such as Apgar scores remained stable, while the survival of infants with lower birth weight, FGR/IUGR, and RDS increased, leading to higher rates of complications such as pROP.

The distribution of *MTHFR* and *PON1* genotypes differed between infants, their mothers, and the reference population. Preterm infants had a lower frequency of carriers of the *MTHFR* rs1801133T allele compared to the general population, which may indicate a selective loss of fetuses carrying this allele under adverse intrauterine conditions. Conversely, homozygotes for the *PON1* rs854560T allele were observed more frequently among preterm infants than in mothers, suggesting a role of the decreased concentration of PON1 in the fetus in susceptibility to preterm birth or adverse neonatal outcomes. No significant differences were found for *PON1* rs662, indicating that this variant has a limited or no contribution to the risk of preterm birth in our population.

We identified several significant associations between infant genotypes and neonatal outcomes. The *MTHFR* rs1801133T allele increased the risk of ELBW (OR = 1.65) with sex-dependent trends: the TT genotype was particularly detrimental in female infants, while the CT genotype conferred risk for both ELBW and ELGA in males. This allele was also associated with increased risks of BPD and PDA. Also, the *PON1* rs662AG genotype showed a borderline association with PDA, suggesting possible involvement of Hcy-TLC-related pathways. Interestingly, carriers of the *MTHFR* rs1801133T allele and *PON1* rs662G allele were more common among infants who died in the early postnatal period. However, statistical significance was not reached, likely due to limited case numbers. No associations were observed with RDS, IVH, NEC, or ROP. Importantly, maternal genotypes were not associated with extreme prematurity, pointing to a predominant role of fetal genetic factors.

Our meta-analysis of five studies, including our own, confirmed the contribution of the neonatal *MTHFR* rs1801133 genotype to the determination of lower birth weight. Homozygous TT carriers had a significantly higher risk of low birth weight, with the TT vs. CT model yielding the most significant association (OR = 1.41; *p* = 0.0097). However, in the recessive model (TT vs. CT + CC), moderate heterogeneity (60%) was observed between populations. The analyses of subgroups revealed trends: in developed countries such as the UK and Canada, the T allele was protective, whereas in China, Turkey, and Poland, it was associated with increased risk. These findings suggest population-specific modifiers influencing the risk. Maternal nutritional status appears to play a crucial role in determining LBW across different countries, while in developed countries the widespread use of folic acid may mitigate the risk associated with *MTHFR* variants.

Our results contrast with earlier meta-analyses, which emphasized the role of maternal rather than infant genotypes. Wu et al. (2017) identified maternal T allele carriers as a risk factor for preterm birth and LBW, particularly in Asian and Caucasian cohorts [10]. Similarly, the most recent meta-analysis by Vafapour et al. (2025) confirmed maternal associations, though limited to Asian and Indian populations [11]. In contrast, our data support the importance of fetal genotypes, particularly in extreme prematurity. These discrepancies may reflect ethnic differences, methodological variation, or the limited availability of neonatal genotype data in prior studies.

Several studies on the neonatal rs1801133 variant and LBW were also excluded from our meta-analysis due to insufficient data. Yang et al. (2010) [49] and Mei et al. (2015) [50] could not be accessed in full text, while Sukla et al. (2013) [51] and Horikoshi et al. (2016) [9] lacked detailed genotype data. Similarly, Kordas et al. (2009) [52] and Obeid et al. (2023) [53] did not report infant genotypes, with the latter explicitly finding no associations. This raises a risk of publication bias. Moreover, none of the included studies distinguished pathological FGR/IUGR from constitutionally small SGA newborns [54]. We observed that ELBW infants were more than twice as likely to be SGA compared to preterm infants with higher birth weight, suggesting that ELBW may be partially attributable to SGA, whereas no such association was found for ELGA, indicating distinct risk profiles for ELGA and ELBW/SGA [26]. Although our meta-analysis is limited to the available data set, it is noteworthy that the results of our subgroup analysis mirror previous reports on maternal *MTHFR* genotype, with differences between developed and developing countries [10].

The observed associations may be explained by several mechanisms. The *MTHFR* rs1801133 variant reduces enzyme activity, leading to hyperhomocysteinemia, which results in endothelial dysfunction and impaired placental perfusion [55]. This may selectively influence fetal survival and growth, consistent with our observation of a lower T allele frequency in surviving preterm infants. For instance, in a Spanish population, the rs1801133T allele was shown to affect fetal growth parameters even under adequate maternal folate intake, with opposing effects depending on fetal sex [56]. Furthermore, animal studies demonstrated that mouse offspring exposed in utero to diet-induced hyperhomocysteinemia exhibited reduced birth weight, increased perinatal mortality, and impaired brain and muscle development [57]. Notably, several of these effects were sex-specific, highlighting differential fetal susceptibility to disturbances in one-carbon metabolism—a pattern consistent with our findings of sex-specific effects of the studied variants. These mechanisms may also contribute to PDA, as previously reported [58], as well as to the risk of BPD, a severe respiratory complication of prematurity. Notably, our finding represents a novel association in the context of BPD. However, previous studies have linked the *MTHFR* rs1801133 variant to the occurrence of childhood asthma, suggesting its broader role in susceptibility to lung diseases [59].

Second, observed sex-specific associations with ELBW and ELGA suggest differential epigenetic regulation, which is supported by recent evidence of sexually dimorphic placental methylation patterns [60] and different susceptibility of male and female fetuses to Hcy-related disturbances in one-carbon metabolism [56,57]. An epigenome-wide study of pregnancy-related methylation patterns has shown that female fetal plasma DNA is generally less methylated compared to male fetal plasma DNA. This pattern was observed in cells derived from the placenta, with much weaker associations observed in cells from cord blood [60]. These findings underscore the potential role of placental methylation in the observed sex differences. Furthermore, the researchers show that some genomic regions of cord blood DNA were more methylated in the group of women who took vitamin supplements, but this effect was not observed in mothers who were carriers of the rs1801133T allele [61]. In another study, the *MTHFR* rs1801133TT genotype was not associated with a higher global plasma DNA methylation ratio. However, pyrosequencing of a selected locus in the *MTHFR* gene showed that DNA in this region was more methylated in TT homozygotes compared with the CC homozygotes, 16.5% vs. 13.8%, respectively [62,63]. These epigenetic effects may also be influenced by environmental factors related to the mother’s health. For example, placental cells from pregnancies affected by gestational obesity were found to contain more methylated DNA than cells from healthy control women [64]. Different methylation patterns and different maternal allele expression ratios were also observed in placental imprinted genes in preeclampsia and placenta accreta spectrum disorders compared with control samples, as well as a correlation between increased oxidative stress and changes in gene methylation was reported [65].

Third, *PON1* variants may influence antioxidant capacity and vulnerability to oxidative stress, a known contributor to PPROM and neonatal morbidity [66,67,68,69]. The borderline associations with PDA, and the literature linking PON1 to reduced antioxidant activity, give some support to this hypothesis.

In addition to *MTHFR*, our previous studies linked infant genotypes in selenoprotein P [70] and angiotensin receptor 1 [71] to ELBW and ELGA, suggesting that extreme prematurity may result from multiple genetic factors acting through oxidative stress and impaired placental vascular regulation.

The strengths of this study include analysis of both infant and maternal genotypes, the use of a reference population, and integration of findings into a meta-analysis. Limitations include modest subgroup sizes, particularly for rare outcomes such as mortality, limited maternal genetic data, and restriction to a single ethnic group. Post hoc power analysis for the study of the Polish population showed 70% power for detecting the association between the *MTHFR* rs1801133T allele and ELBW (dominant model) and approximately 55% power for comparing the frequency of carriers of this allele between preterm infants and the population sample, indicating moderate statistical power for detecting these major effects. The meta-analysis may, on the other hand, be affected by publication bias, as studies without significant results are less likely to be published. Finally, methodological variability across studies, including inconsistent definitions of SGA and LBW, complicates interpretation.

Our findings suggest that the infant, rather than the maternal, *MTHFR* genotype plays an important role in susceptibility to extreme prematurity and its complications. They also highlight possible sex-specific vulnerability, which warrants further research. Future studies should include larger, ethnically diverse cohorts, rigorous phenotyping distinguishing pathological FGR/IUGR from constitutional SGA, and integration of genetic, epigenetic, and biochemical markers.

## 5. Conclusions

In summary, we provide novel evidence that the fetal *MTHFR* rs1801133 genotype contributes to the risk of ELBW and BPD and extend existing data regarding PDA and perinatal mortality. This is the first study to demonstrate such associations in ELBW infants and to integrate them within a broader meta-analysis. Our findings emphasize the need for comprehensive, multifactorial models of extreme prematurity that combine genetic, environmental, and epigenetic factors and point toward the potential of genetic profiling in risk stratification for the most vulnerable neonatal populations.

## Figures and Tables

**Figure 1 genes-16-01192-f001:**
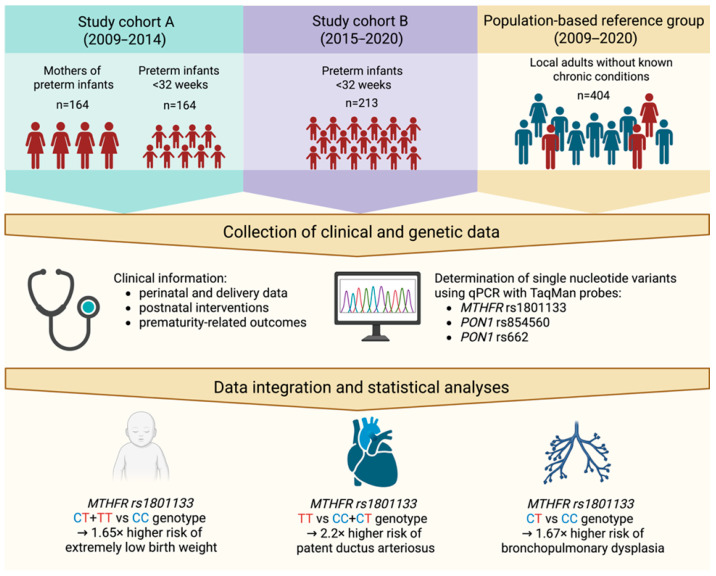
Schematic overview of the study design and obtained results. Created in BioRender. Skulimowski, B. (2025) https://BioRender.com/3l5efzr (accessed on 19 August 2025).

**Figure 2 genes-16-01192-f002:**
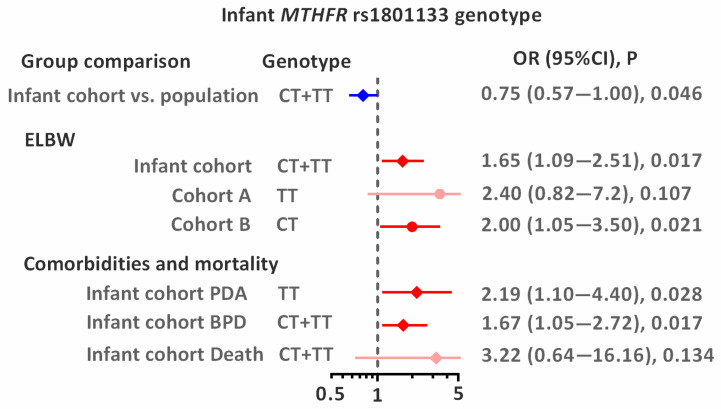
Associations between infant *MTHFR* rs1801133 genotype and studied outcomes. Odds ratios (ORs) with 95% confidence intervals (CIs) are shown. The dashed vertical line indicates the null value (OR = 1), and horizontal lines represent 95% CIs. Blue indicates lower risk, whereas red indicates higher risk (significant association or statistical trend).

**Figure 3 genes-16-01192-f003:**
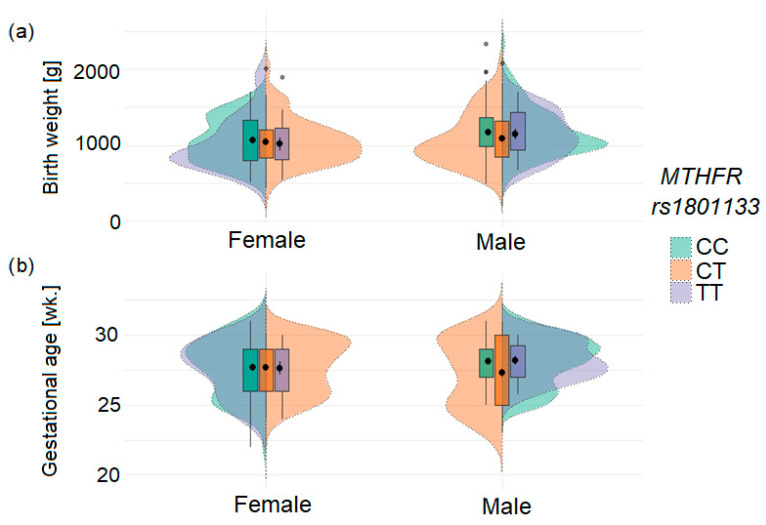
Distribution of birth weight (**a**) and gestational age (**b**) in infants according to sex and *MTHFR* rs1801133 genotype. The plots illustrate differential effects of TT homozygotes and CT heterozygotes by sex. Black dots indicate mean values, and error bars represent standard deviations; grey dots represent outlier values.

**Figure 4 genes-16-01192-f004:**
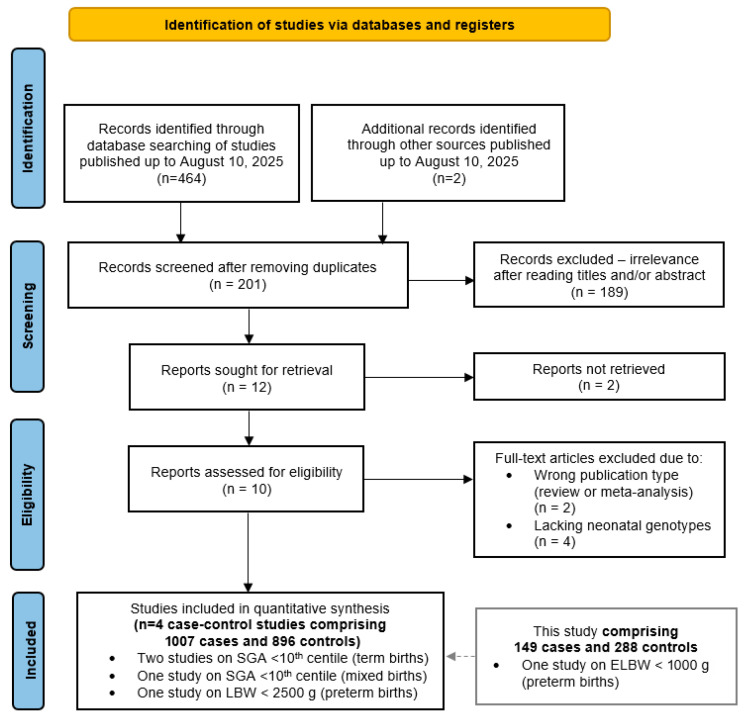
PRISMA diagram showing the flow of study selection for inclusion in the meta-analysis of the *MTHFR* rs1801133 variant and adverse neonatal outcomes.

**Figure 5 genes-16-01192-f005:**
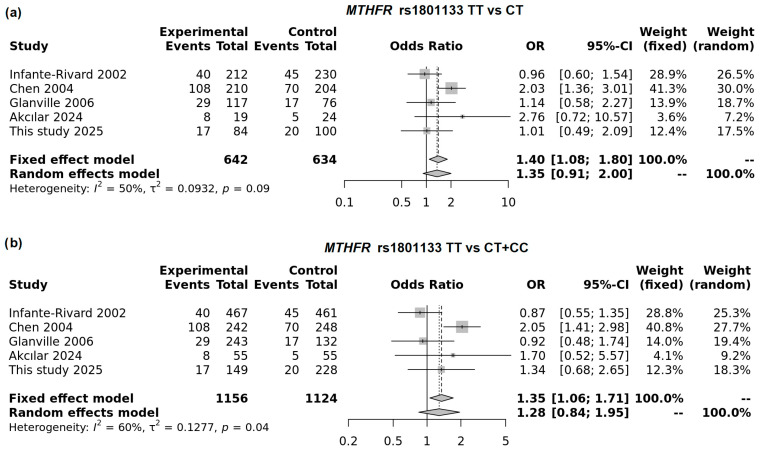
Forest plot on the association of the *MTHFR* rs1801133T allele with LBW in TT vs. CT model (**a**) and recessive model (**b**).

**Figure 6 genes-16-01192-f006:**
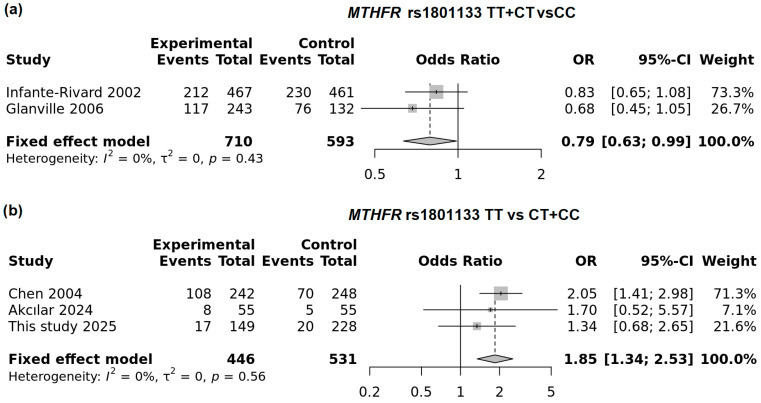
Subgroup analysis of the neonatal *MTHFR* rs1801133T allele: (**a**) dominant model in developed countries and (**b**) recessive model in other countries (including this study).

**Figure 7 genes-16-01192-f007:**
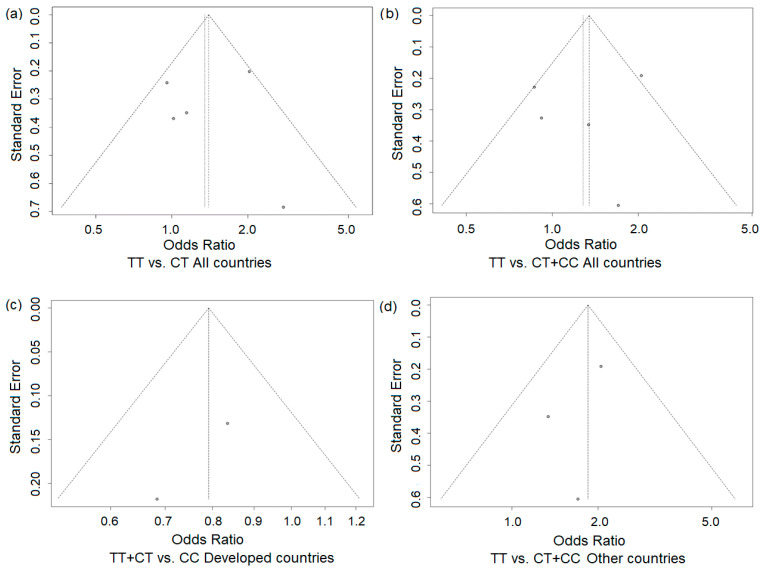
Funnel plots for the association of the *MTHFR* rs1801133 T allele across genetic models and study populations: all countries (**a**,**b**) and developed (Canada and the UK (**c**)) and other countries (**d**). Each dot represents an individual study included in the meta-analysis.

**Table 1 genes-16-01192-t001:** Demographic and clinical characteristics of premature infants and their associations with ELBW and ELGA.

Parameter		All Infants *n* = 377	Infants with ELBW; *n* = 149	Infants with ELGA; *n* = 152
Gestational age [weeks]	Mean (SD)	27.8 (2.1)	26.3 (1.9) ****	25.7 (1.1) ****
	Range	22.0–31.9	22–30.0	22–27.0
	<28 weeks, *n* (%)	152 (40.3)	111 (74.5) ****	152 (100.0)
Birth weight [g]	Mean (SD)	1101.6 (321.5)	801.9 (123.8) ****	896.1 (217.9) ****
	Range	432–2340	432–995	432–1780
	<1000 g, *n* (%)	149 (39.5)	149 (100.0)	111 (73.0) ****
	SGA (<10th), *n* (%)	99 (26.3)	54 (36.2) ***	17 (11.2) ****
	FGR/IUGR, *n* (%)	25 (6.6)	22 (14.8) ****	5 (3.6)
Male sex, *n* (%)		206 (54.6)	74 (49.7)	81 (53.0)
Risk factors at birth				
PPROM, *n* (%)		107 (28.4)	40 (26.8)	50 (32.9)
PPROM > 3 d, *n* (%)		74 (19.6)	23 (15.1)	31 (20.4)
Delivery by cesarean section, *n* (%)		204 (54.1)	74 (49.7)	68 (44.7) **
Apgar 1, Median (Q1, Q3)		5 (2, 7)	3 (2, 6) ****	4 (2, 6) ****
Apgar 5, Median (Q1, Q3)		7 (6, 8)	7 (5.5, 7) ****	7 (6, 7) ****
Parameters related to respiratory failure				
Surfactant treatment, *n* (%)		181 (48.0)	94 (63.1) ****	91 (59.9) ****
Resuscitation, *n* (%)		317 (84.1)	141 (94.6) ****	141 (92.8) ***
Mech. vent., *n* (%)		258 (68.4)	133 (89.3) ****	139 (91.4) ****
Mech. vent. period [d], Median (Q1, Q3)		13 (2, 33)	33 (17, 25) ****	35.5 (18, 53) ****
O_2_ supply period [d], Median (Q1, Q3)		28 (1, 58)	59 (30, 75) ****	58.5 (27.5, 74) ****
Blood transfusions, Median (Q1, Q3)		3 (1, 6)	6 (3, 8) ****	5.5 (3, 8) ****
Complications of prematurity, *n* (%)				
NEC		79 (21.0)	53 (35.6) ****	52 (34.2) ****
BPD		149 (39.5)	113 (75.8) ****	112 (73.7) ****
IVH		218 (57.8)	112 (75.2) ****	116 (76.3) ****
RDS		251 (66.5)	115 (77.2) ***	113 (74.1) **
PVL		28 (7.4)	18 (12.1) **	17 (11.2) *
ROP		239 (63.3)	129 (86.6) ****	136 (89.5) ****
PDA		104 (27.6)	55 (36.9) **	49 (32.2)
Death		8 (2.1)	7 (4.7) **	7 (4.6) **

Abbreviations and symbols: BPD—bronchopulmonary dysplasia; FGR—fetal growth restriction; IUGR—intrauterine growth retardation; IVH—intraventricular hemorrhage; NEC—necrotizing enterocolitis; PDA—patent ductus arteriosus; PPROM—preterm prelabor rupture of membranes; PVL—periventricular leukomalacia; RDS—respiratory distress syndrome; ROP—retinopathy of prematurity; SGA—small for gestational age. Statistical significance: * *p* < 0.05; ** *p* ≤ 0.01; *** *p* ≤ 0.001; **** *p* ≤ 0.0001.

**Table 2 genes-16-01192-t002:** Comparison of *MTHFR* and *PON1* genotype frequencies between study groups.

Genotype and Allele Frequencies	Population *n* = 404	Mothers (Cohort A) *n* = 164	Infants (Cohort A) *n* = 164	All Infants *n* = 377
*MTHFR* rs1801133C>T (Ala222Val)				
*CC*	178 (44.1)	80 (48.8)	86 (52.4)	193 (51.2)
*CT*	185 (45.8)	69 (42.1)	62 (37.8)	147 (39.0)
*TT*	41 (10.1)	15 (9.1)	16 (9.8)	37 (9.8)
MAF; *P_HWE_*	0.330; 0.081	0.302; 0.982	0.287; 0.627	0.293; 0.903
*PON1* rs854560A>T (Leu55Met)				
*AA*	159 (39.4)	74 (45.1)	67 (40.9)	146 (39.0)
*AT*	194 (48.0)	77 (47.0)	76 (46.3)	173 (46.3)
*TT*	51 (12.6)	13 (7.9)	21 (12.8)	55 (14.7)
MAF; *P_HWE_*	0.366; 0.490	0.314; 0.250	0.360; 0.997	0.378; 0.748
*PON1* rs662A>G (Gln192Arg)				
*AA*	199 (49.3)	85 (51.8)	77 (47.0)	191 (50.9)
*AG*	178 (44.1)	71 (43.3)	78 (47.6)	160 (42.7)
*GG*	27 (6.7)	8 (4.9)	10 (6.1)	24 (6.4)
MAF; *P_HWE_*	0.287; 0.125	0.265; 0.156	0.297; 0.237	0.277; 0.212
Statistical analysis				
Variant	Genetic model	Comparison	OR (95% CI); *p*-value	
rs1801133	Dominant (CT + TT)	All infants vs. Population	**0.75 (0.57–0.995); 0.046**	
rs854560	Recessive (TT)	All infants vs. Mothers	**2.00 (1.06–3.78); 0.029**	

Abbreviations and symbols: MAF (minor allele frequency) and P_HWE_ (*p*-value for Hardy–Weinberg equilibrium). Statistically significant values are highlighted in bold.

**Table 3 genes-16-01192-t003:** Distribution of studied genetic variants in preterm infants and their mothers from the Polish population in relation to time of enrollment, extremely low birth weight (ELBW) and extremely low gestational age (ELGA). The table presents Hardy–Weinberg equilibrium (HWE) *p*-values, minor allele frequencies (MAFs), and the impact of SNVs on ELBW and ELGA.

Genotype and Allele Frequencies	Study Cohort A (2009–2014)								Study Cohort B (2015–2020)			
	Mothers; *n* = 164				Infants; *n* = 164				Infants; *n* = 213			
	ELBW		ELGA		ELBW		ELGA		ELBW		ELGA	
	No; *n* = 110	Yes; *n* = 54	No; *n* = 96	Yes; *n* = 68	No; *n* = 110	Yes; *n* = 54	No; *n* = 97	Yes; *n* = 67	No; *n* = 118	Yes; *n* = 95	No; *n* = 128	Yes; *n* = 85
*MTHFR* rs1801133C>T (Ala222Val)												
CC	53 (48.2)	27 (50.0)	45 (46.9)	35 (51.5)	61 (55.5)	25 (46.3)	54 (55.7)	32 (47.8)	67 (56.8)	40 (42.1)	68 (53.1)	39 (45.9)
CT	47 (42.7)	22 (40.7)	42 (43.8)	28 (41.2)	41 (37.3)	21 (38.9)	33 (34.0)	29 (43.3)	39 (33.1)	46 (48.4)	46 (35.9)	39 (45.9)
TT	10 (9.1)	5 (9.3)	9 (9.4)	5 (7.4)	8 (7.3)	8 (14.8)	10 (10.3)	6 (9.0)	12 (10.2)	9 (9.5)	14 (10.9)	7 (8.2)
MAF	0.305	0.296	0.313	0.279	0.259	0.343	0.273	0.306	0.267	0.337	0.289	0.312
PHWE	0.927	0.866	0.859	0.852	0.760	0.315	0.158	0.876	0.091	0.414	0.155	0.523
*PON1* rs854560A>T (Leu55Met)												
AA	49 (44.5)	25 (46.3)	41 (42.7)	33 (48.5)	45 (40.5)	22 (41.5)	39 (40.6)	28 (41.2)	42 (36.5)	37 (38.9)	48 (38.1)	31 (36.9)
AT	53 (48.2)	24 (44.4)	47 (49)	30 (44.1)	49 (44.1)	27 (50.9)	46 (47.9)	30 (44.1)	55 (47.8)	42 (44.2)	57 (45.2)	40 (47.6)
TT	8 (7.3)	5 (9.3)	8 (8.3)	5 (7.4)	17 (15.3)	4 (7.5)	11 (11.5)	10 (14.7)	18 (15.7)	16 (16.8)	21 (16.7)	13 (15.5)
MAF	0.314	0.315	0.328	0.294	0.374	0.330	0.354	0.368	0.396	0.389	0.393	0.393
PHWE	0.212	0.824	0.280	0.606	0.547	0.269	0.642	0.673	0.999	0.493	0.562	0.987
*PON1* rs662A>G (Gln192Arg)												
AA	57 (51.8)	28 (51.9)	47 (49)	38 (55.9)	55 (49.1)	22 (41.5)	45 (46.4)	32 (47.0)	63 (54.8)	51 (53.7)	69 (55.2)	45 (52.9)
AG	50 (45.5)	21 (38.9)	45 (46.9)	26 (38.2)	50 (44.6)	28 (52.8)	48 (49.5)	30 (44.1)	43 (37.4)	39 (41.4)	45 (36.0)	37 (43.5)
GG	3 (2.7)	5 (9.3)	4 (4.2)	4 (5.9)	7 (6.3)	3 (5.7)	4 (4.1)	6 (8.8)	9 (7.8)	5 (5.3)	11 (8.8)	2 (3.5)
MAF	0.255	0.287	0.276	0.250	0.286	0.321	0.289	0.309	0.265	0.258	0.268	0.253
PHWE	0.038	0.714	0.090	0.872	0.321	0.122	0.044	0.783	0.091	0.478	0.357	0.076
Statistical analysis												
Studied variant		Allele model or Genotype			Comparison: Yes vs. No				OR (95% CI); *p*-value			
rs1801133		CT + TT vs. CC (Dominant)			ELBW: Infants Cohort A + B				1.65 (1.09–2.51); 0.017			
rs1801133		TT vs. CC			ELBW: Infants Cohort A				2.40 (0.82–7.2); 0.107			
rs1801133		CT + TT vs. CC (Dominant)			ELBW: Infants Cohort B				1.81 (1.05–3.12), 0.033			
rs1801133		CT vs. CC			ELBW: Infants Cohort B				2.00 (1.05–3.50), 0.021			
rs1801133		CT vs. CC			ELGA: Infants Cohort A				1.50 (0.76–2.90); 0.244			
rs1801133		CT vs. CC			ELGA: Infants Cohort B				1.50 (0.83–2.60); 0.187			

**Table 4 genes-16-01192-t004:** Distribution of studied genetic variants in a cohort of 377 preterm infants in relation to comorbidities and mortality.

Genotype and Allele Frequencies	RDS		IVH		PDA		NEC		BPD		ROP		pROP		Death	
	No *n* = 126	Yes *n* = 251	No *n* = 159	Yes *n* = 218	No *n* = 273	Yes *n* = 104	No *n* = 298	Yes *n* = 79	No *n* = 228	Yes *n* = 149	No *n* = 137	Yes *n* = 239	No *n* = 257	Yes *n* = 120	No *n* = 369	Yes *n* = 8
*MTHFR* rs1801133C>T (Ala222Val)																
CC	50.8	51.4	52.2	50.5	51.1	50.0	51.7	49.4	55.3	45.0	54.0	50.6	54.5	45.8	51.8	25.0
CT	41.3	37.8	38.4	39.4	40.2	34.6	39.3	38.0	33.8	47.0	36.5	38.5	35.0	44.2	38.5	62.5
TT	7.9	10.8	9.4	10.1	7.6	15.4	9.1	12.7	11.4	8.1	9.5	10.9	10.5	10.0	9.8	12.5
MAF	0.286	0.297	0.286	0.298	0.277	0.327	0.287	0.316	0.279	0.315	0.278	0.302	0.280	0.321	0.290	0.438
*PON1* rs854560A>T (Leu55Met)																
AA	45.5	35.9	39.1	39.0	35.9	47.1	40.0	35.4	40.4	36.9	37.8	39.7	38.3	41.8	38.8	50.0
AT	44.7	47.0	48.1	45.0	48.5	40.4	47.1	43.0	47.1	45.0	47.4	45.6	47.5	41.8	46.2	50.0
TT	9.8	17.1	12.8	16.1	15.6	12.5	12.9	21.5	12.4	18.1	14.8	14.6	14.2	16.5	15.0	0.0
MAF	0.321	0.406	0.369	0.385	0.398	0.327	0.364	0.430	0.360	0.406	0.385	0.374	0.380	0.373	0.381	0.250
*PON1* rs662A>G (Gln192Arg)																
AA	53.2	49.8	54.8	48.2	50.1	36.8	50.7	51.9	47.8	55.7	50.7	51.0	52.7	44.3	51.5	25.0
AG	41.1	43.4	39.5	45.0	42.0	55.6	42.9	41.8	44.7	39.6	44.1	41.8	41.2	48.1	42.0	75.0
GG	5.6	6.8	5.7	6.9	7.0	7.5	6.4	6.3	7.5	4.7	5.1	7.1	6.1	7.6	6.5	0.0
MAF	0.262	0.285	0.255	0.294	0.280	0.354	0.279	0.272	0.299	0.245	0.272	0.280	0.267	0.316	0.275	0.375
Statistical analysis																
Studied variant		Genetic model/Genotype			Comparison Yes vs. No						OR (95% CI); *p*-value					
rs1801133		Recessive			PDA						2.19 (1.10–4.40); 0.028					
rs1801133		CT vs. CC			BPD						1.67 (1.05–2.72); 0.017					
rs1801133		Dominant			Death						3.22 (0.64–16.2); 0.156					
rs662		AG vs. AA			PDA						1.80 (0.99–3.30); 0.053					
rs662		Dominant			Death						3.19 (0.65–16.4); 0.151					

Abbreviations and symbols: BPD—bronchopulmonary dysplasia; IVH—intraventricular hemorrhage; MAF—minor allele frequency; NEC—necrotizing enterocolitis; PDA—patent ductus arteriosus; pROP—proliferative ROP; RDS—respiratory distress syndrome; ROP—retinopathy of prematurity.

**Table 5 genes-16-01192-t005:** Characteristics of studies included in the meta-analysis of the association between neonatal *MTHFR* rs1801133 genotype and LBW.

First Author and Year	Country (Ethnicity)	Study Includes Preterm	Outcome	Method	Cases/ Controls	Cases			Controls			WG MAFs	Controls P_HWE_	Q Score (NOS)
						CC	CT	TT	CC	CT	TT			
Infante-Rivard 2002 [38]	Canada (Mixed)	mixed	SGA	PCR	467/461	255	172	40	231	185	45	0.284	0.375	8
Chen 2004 [39]	China (Asian)	yes	LBW	PCR	242/248	32	102	108	44	134	70	0.604	0.145	6
Glanville 2006 [40]	UK (Caucasian)	no	SGA	PCR-RFLP	243/132	126	88	29	56	59	17	0.319	0.813	6
Akcılar 2024 [41]	Turkey (Caucasian)	no	SGA	PCR-RFLP	55/55	36	11	8	31	19	5	0.255	0.414	6
This study 2025	Poland (Caucasian)	yes	ELBW	TaqMan	149/228	65	67	17	128	80	20	0.293	0.151	8
Total					1156/1124	514	440	202	490	477	157	---	---	---

Abbreviations: ELBW—extremely low birth weight (<1000 g); LBW—low birth weight; MAF—minor allele frequency; NOS—Newcastle–Ottawa Score; PCR—polymerase chain reaction; PCR-RFLP—polymerase chain reaction–restriction fragment length polymorphism; P_HWE_—statistical significance for comparison between observed genotype frequencies and those expected based on Hardy–Weinberg equilibrium; Q—quality; RT-PCR—real-time polymerase chain reaction; SGA—small for gestational age (<10th centile); WG—whole group.

## Data Availability

The original contributions presented in this study are included in the article/Appendix A. Further inquiries can be directed to the corresponding author.

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
