# Peer review of "Newborn MTHFR rs1801133 Variant and Extremely Low Birth Weight: A Case–Control Study and Meta-Analysis"

_genes, 2025, doi:10.3390/genes16101192_

Round 1
Reviewer 1 Report
Comments and Suggestions for Authors
The authors studied the association of MTHFR rs 1801133 variant and low birth weight and presented its novel finding of increased bronchopulmonary dysplasia risk amongst others.
The research was methodically carried out and carefully interpreted and presented. The interaction of genetic make-up (Caucasian) and possible social (folate supplement) is interesting to the readers.
Line 256 contains a typographical error.
Reviewer 2 Report
Comments and Suggestions for Authors
Comments on „Newborn MTHR rs1801133 Variant and Extremely Low birth weight: a case control study and meta-analysis” Skulimowski et al. 2025
The manuscript presents a statical meta-analysis of MTHR variants in newborns where they are associated with low birth weight and prematurity. The authors did an excellent job collecting data and doing statical analysis. I personally do not have any major comments.
Introduction – Superb! The introduction is short but very comprehensive and informative. The authors do not waste any time getting into details for the MTHFR and PON1 genes and the reasons for the meta-analysis.
Lines 54-58 – could the authors give one or more examples of other genes (besides MTHFR) that affect birth weight?
Lines 85-89 – the purpose of the manuscript is very well defined. Well done.
Materials and Methods – excellent, I do not have any major comments. The authors explained very well their study criteria, data collection and statistical analysis. The section of the manuscript is in y opinion very complete.
Line 101 – are the infants Caucasian because of the population in the region, or was it a criterion for the study?
Line 194 – what do the authors mean by “… accessed 11 September 2025.”? The date is after submission. I can be corrected if I misunderstood.
Results – The result section is in my opinion rather complete and the authors go in detail the statical analysis of their research.
Table 1 – very good. No comments.
Table 2 and Figure 2 – No comments.
Table 3, Figure 3, Table 4, Table/Figure5 – well done.
Discussion – No comments.
Reviewer 3 Report
Comments and Suggestions for Authors
The authors describe the correlation of the MTHFR rs1801133 variant with extremely low birth weight (ELBW) in a cohort of premature newborns. They also report associations of this variant with bronchopulmonary dysplasia, patent ductus arteriosus, and mortality. The study is well-conducted, and the data are comprehensively reported. I have no concerns regarding the methodology or results.